# Zika virus isolation, propagation, and quantification using multiple methods

Worawat Dangsagul[1], Kriengsak Ruchusatsawat[2], Apiwat Tawatsin[2], Don Changsom[2], Pirom Noisumdaeng[3,4], Sukontip Putchakarn[2], Chayawat Phatihattakorn[5], Prasert Auewarakul[5], Pilaipan Puthavathana[1]*

1 Faculty of Medical Technology, Mahidol University, Nakhon Pathom, Thailand, 2 Department of Medical Sciences, Ministry of Public Health, Nonthaburi, Thailand, 3 Faculty of Public Health, Thammasat University, Pathum Thani, Thailand, 4 Thammasat University Research Unit in Modern Microbiology and Public Health Genomics, Thammasat University, Pathum Thani, Thailand, 5 Faculty of Medicine Siriraj Hospital, Mahidol University, Bangkok, Thailand

* pilaipan.put@mahidol.edu

## Abstract

Zika virus (ZIKV) was isolated from the archival urine, serum, and autopsy specimens by intrathoracic inoculation of *Toxorhynchitis splendens* and followed by three blind sub-passaging in C6/36 mosquito cells. The virus isolates were identified using an immunofluorescence assay and real-time reverse transcription-polymerase chain reaction (real-time RT-PCR). This study analyzed 11 ZIKV isolates. One isolate (0.6%) was obtained from 171 urine samples, eight (8.7%) from 92 serum samples and two from tissues of an abortive fetus. After propagation in C6/36 cells, ZIKV was titrated by plaque and focus forming unit (FFU) assays in Vero cell monolayers, and viral genomes were determined via real-time and digital RT-PCR. Plaque and FFU assay quantitations were comparable, with the amount of infectious viruses averaging $10^6$–$10^7$ PFU or FFU/ml. Real-time RT-PCR semi-quantified the viral genome numbers, with Ct values varying from 12 to 14. Digital RT-PCR, which precisely determines the numbers of the viral genomes, consistently averaged 10–100 times higher than the number of infectious units. There was good correlation between the results of these titration methods. Therefore, the selection of a method should be based on the objectives of each research studies.

## Introduction

Zika virus (ZIKV) was first isolated from a captive rhesus monkey on a canopy in Uganda during surveillance for Yellow fever in 1947 [1]. Subsequently, Zika fever was found in three human cases in Nigeria [2]. The first documented outbreak of Zika fever was in 2007 on the Western Pacific island of Yap in the Federated States of Micronesia [3]. This was followed by an epidemic in French Polynesia in the South Pacific in 2013 and 2014, which had an unusual number of cases complicated with Guillain-Barre syndrome [4]. In March 2015, ZIKV emerged in Brazil and included the highest incidence of congenital microcephaly ever reported [5].

**Funding:** This study was financially supported by the National Science and Technology Development

Agency, Thailand, Grant No. P-17-50551. The funders had no role in study design, data collection and analysis, decision to publish, or preparation of the manuscript.

**Competing interests:** The authors have declared that no competing interests exist.

ZIKV is a risk group 2 agent. Therefore, study of this infectious virus can be conducted in a biosafety laboratory level 2. It is fastidious and difficult to isolate, with rates lower than other emerging viruses [6]. The first ZIKV isolate was, unintentionally, discovered by intracerebral inoculation of suckling mice with a serum specimen from a sentinel rhesus monkey and identified by a neutralization test [1]. Later, several attempts were made to isolate the ZIKV in multiple host systems, including those described for flaviviruses, e.g., the *Aedes albopictus* and *Toxorhynchytes splendens* mosquitoes. Subsequently the C6/36 cell line derived from *Ae. albopictus*, and the Vero cell line (ATCC CCL-81) derived from African green monkey kidney were also assessed [7–10]. The success rates of ZIKV isolation in different cell lines are not significantly different [7].

Viral quantification is fundamental to establishing a standard protocol which makes the results from multiple laboratories comparable. The numbers of viruses in a preparation can be determined by approaches which may vary depending on the objectives of the experiment or study. Molecular techniques, such as real-time reverse transcription-polymerase chain reaction (RT-PCR), yield the cycle threshold (Ct) which is a rough estimate of the number of the viral genomes in a virus preparation [11]. At present, several advanced molecular techniques can enumerate the copy numbers of the viral genomes with high precision, such as quantitative real-time PCR [12] and digital droplet PCR [13]. Even though these molecular techniques are highly sensitive, they cannot differentiate between the genomes derived from infectious and dead viruses. In this report we share our experience on the isolation of ZIKV using *Tx. splendens* mosquitoes in adjunction with C6/36 cells. The ZIKV isolates were quantified for infectious viruses by plaque assay and FFU assay, and for viral genomes by real-time RT-PCR and digital droplet RT-PCR.

## Materials and methods

### Ethics statement

Research use of human samples received approval from the Mahidol University Central Institutional Review Board (MU-CIRB): protocol number MU-CIRB 2017/180.1210. Anonymized archival serum and urine samples for ZIKV isolation were kindly provided by the Regional Medical Sciences Centers and the National Institute of Health (NIH), Department of Medical Sciences, Ministry of Public Health. They were the leftover specimens after laboratory diagnostic testing for ZIKV infection between 2016 and 2018. The archival autopsy tissues from an abortive fetus with congenital microcephaly in 2016 were obtained from the Faculty of Medicine Siriraj Hospital, Mahidol University. The IRB waived the requirement for informed consent for this study.

Use of mosquitoes in the experiments was approved by the Mahidol University-Institutional Animal Care and Use Committee (number MUVS-2018-01-01).

### ZIKV isolation in mosquitoes

A colony of *Tx. splendens* maintained at the Medical Insect Unit, NIH was used in this study. Mosquito larvae were fed with *Aedes* mosquito larvae, and the adults were fed with 5% sugar syrup supplemented with 10% multi-vitamins. Each of a group of 10 mosquitoes of age 5–10 days, was inoculated intrathoracically with approximately 0.3 µl of a clinical specimen, and a group of 5 un-inoculated mosquitoes was kept as the negative control in every mosquito inoculation experiment. The inoculated mosquitoes were kept for 10 days before harvesting. Pool of 10 mosquitoes was ground and suspended in L-15 media (Gibco, MA) supplemented with 10% FBS (Gibco). Mosquito suspensions were centrifuged at 5,000 rpm for 30 minutes at 4˚C,

then filtered with a 0.4 μm filtered membrane, and subjected to the viral genome detection by real-time RT-PCR targeting the NS2B region of the ZIKV genomes.

## ZIKV propagation in C6/36 cells

The C6/36 cell line (ATCC CRL-1660) is derived from *Ae. albopictus* mosquitoes and grown in Leibovitz's-15 (L-15) medium (Life Technologies, NY) supplement with 10% fetal bovine serum (FBS) (Life Technologies), 10% tryptose phosphate broth, and antibiotics at 28˚C in the absence of $CO_2$. A mosquito suspension positive for ZIKV genomes was inoculated onto the C6/36 cell monolayers and maintained in medium supplemented with 5% FBS. Viral propagation was carried out through three blinded passages; each passage took about 10 days. At the end of each passage, the cell monolayer was scraped off of the plastic surface and tested for the flaviviral antigen by indirect immunofluorescence assay (IFA) using the 4G2 monoclonal antibody specific to flavivirus E antigen. In parallel, ZIKV in the culture supernatants were identified by real-time RT-PCR. The ZIKV isolates were propagated in C6/36 cells; culture supernatants were spun, aliquot and kept as the virus stocks at -70˚C until used.

## Real time RT-PCR

Total RNA was extracted from virus suspensions using the Qiagen Viral RNA Mini Kit (Qian, Hilden, Germany). A reaction of 25 μl volume was set up comprising 5 μl of RNA extract, 12.5 μl of 2X reaction buffer (Superscript® III one-step RT-PCR system with Platinum® Taq polymerase) (Thermo Fisher Scientific, Darmstadt, Germany, 0.5 μl of primers and probes, 1 μl of SuperScript® III/Platinum® *Taq* Mix and 5 μl of deionized distilled water. The primers and probe targeting the NS2B region of the ZIKV genome were designed by the Pan American Health Organization (PAHO) [14], with the Zika 4481 forward primer sequence: 5'-CTGT GGCATGAACCCAATAG-3'; and the Zika 4552c reverse primer sequence: 5'-ATCCCATAGA GCACCACTCC-3'; and the Zika 4507c probe sequence: 5'- CCACGCTCCAGCTGCAAAG G-3'. Reaction cycles started with a reverse transcription step at 50˚C for 30 minutes, followed by a step of inactivation of reverse transcriptase and activation of DNA polymerase at 95˚C for 10 minutes, and 45 cycles of amplification comprised of DNA denaturation for 15 seconds at 95˚C, and the DNA annealing and extension for 1 minute at 55˚C.

## Virus titration

In general, virus quantitation can be accomplished by many means, depending on the purpose of its further use. Herein, the ZIKV stock was titrated for the numbers of infectious viruses by plaque assay and the focus-forming unit (FFU) assay in Vero cell monolayers. The numbers of the viral genomes were quantitated by real-time RT-PCR and droplet digital RT-PCR.

**Plaque assay.** Vero cells (ATCC CCL-81) derived from an African green monkey kidney were grown in Eagle's minimum essential medium (EMEM) (Gibco, MA) supplement with 10% FBS and maintained in media supplemented with 2% FBS at 37˚C in a $CO_2$ incubator. The ZIKV stock was 10 fold-serially diluted, and each dilution was inoculated onto the Vero cell monolayers in 12-well culture plates in triplicate. Virus inoculums were allowed to absorb onto the cell surface for 2 hours at 37˚C before discarding. The culture plates were overlaid with a semisolid media containing 1X MEM, 2% FBS, and 1.5% carboxymethyl cellulose and further incubated for 7 days. The culture plates were then fixed with 3% formaldehyde for 1 hour before discarding the overlay media. Plates were washed with water and stained with 1% crystal violet for 15 minutes, then washed again. Plaques (foci of infected cells) were counted, and viral titers were calculated and expressed in terms of the plaque-forming unit (PFU)/ml. Plaque sizes were measured by digital Vernier caliper.

**Focus forming unit assay.** The ZIKV stock was 10-fold serially diluted, and each dilution was inoculated onto Vero cell monolayers in 96-well culture plates in triplicate. Virus inoculums were allowed to absorb onto the cell monolayers for 2 hours at 37°C before discarding. The culture plates were overlaid with a semisolid media containing 1X MEM, 2% FBS, and 1.5% carboxymethyl cellulose. Three days-post-infection; the culture plates were fixed with 3% formaldehyde and permeabilized with 1% Triton X-100. The plates were then stained with a 4G2 monoclonal antibody followed by a goat anti-mouse IgG conjugated with horseradish peroxidase (Southern Biotech, Atlanta, GA). The chromogenic substrate 3, 3'-diaminobenzidine tetrahydrochloride was added, producing dark brown foci in antigen-positive cells. Due to the tiny sizes of these foci of infected cells, we photographed each well of the reaction plates and magnified the wells using Icy software to facilitate the counting of foci [15]. Two scientists independently performed the counting. The viral titers were presented as foci forming units (FFUs)/ml.

**Droplet digital RT-PCR.** Total RNA extracted from the culture supernatants was analyzed for the number of ZIKV genomes by the droplet digital real-time RT-PCR using One-Step RT-ddPCR Advanced Kit for Probes (Bio-Rad, Hercules, CA). The reaction employed the same primers and probe targeting the NS2B region that was used for the real-time RT-PCR. A reaction volume of 22 μl consisted of 5.5 μl of the RNA extract and 16.5 μl of the master mix containing 1X Supermix, 20 U/μl reverse transcriptase, 15 mM DTT, 500 nM forward primer, 900 nM reverse primer, and 200 nM probe. A 20 μl volume of the reaction mixture was transferred to DG8™ cartridges (Bio-Rad) and 70 μl of Droplet Generation Oil for Probes (Bio-Rad) was added. Droplets were generated with a QX200™ Droplet Generator (Bio-Rad), and a 40 μl volume of the droplet suspension was processed with the QX200™ Droplet Digital™ PCR system (Bio-Rad) in a T100™ Thermal Cycler (Bio-Rad).

The thermal cycling conditions were comprised of these steps: reverse transcription at 50°C for 60 minutes; inactivation of reverse transcriptase and activation of DNA polymerase at 95°C for 10 minutes; DNA amplification reaction which employed 50 cycles of DNA denaturation at 95°C for 30 seconds, DNA annealing and elongation at 55°C for 60 seconds; DNA polymerase inactivation at 98°C for 10 minutes. According to the manufacturer, a maximum of 20,000 oil droplets are generated in a reaction, and each droplet can accommodate only one RNA target for an amplification reaction. The amplified DNA products were detected and enumerated by the Bio-Rad QX200 Droplet Reader and QuantaSoft™ software version 1.7.4 (Bio-Rad), respectively. The threshold to separate the clusters of droplets containing the target RNA amplification from that of the non-template negative control was set manually across the entire reaction plate.

**Genomic characteristics of the ZIKV isolates.** Complete genome sequencing by the Sanger method showed that all 11 ZIKV isolates belonged to the Asian lineage. These sequences can be retrieved from the GenBank database using the accession numbers shown in Table 1.

## Results

### Isolation of ZIKV

A total of 270 archival specimens (92 serum, 171 urine, and 7 autopsy tissue samples) from ZIKV-infected individuals were inoculated into *Tx. splendens* mosquitoes. Of 270 pooled mosquito suspensions, 30 were positive for the viral genomes as detected by real-time RT-PCR. These 30 suspensions were inoculated onto C6/36 cell monolayers, but only 11 virus isolates were obtained after three blinded subpassages. The results are shown in Table 1.

**Table 1. Plaque size of Zika virus isolates in 1.5% CMC.**

| Virus name | Passage history | Specimen type | Plaque sizes (mm.) ($\bar{X}$, sd, min-max) | Accession No. |
|---|---|---|---|---|
| MUMT-1/2016 | TS-1, C6/36-4 | Brain stem | 3.49, 0.55, 2.37–4.40 | MT377492 |
| MU-DMSC-2/2016 | TS-1, C6/36-3 | Serum | 1.13, 0.24, 0.44–1.94 | MT377496 |
| MU-DMSC-3/2016 | TS-1, C6/36-3 | Urine | 1.24, 0.24, 0.48–1.64 | MT377497 |
| MU-DMSC-4/2016 | TS-1, C6/36-4 | Serum | 2.18, 0.59, 1.45–3.37 | MT377498 |
| MU-DMSC-5/2016 | TS-1, C6/36-3 | Serum | 1.59, 0.54, 0.90–3.18 | MT377499 |
| MU-DMSC-6/2016 | TS-1, C6/36-3 | Serum | 0.79, 0.58, 0.17–2.24 | MT377500 |
| MU-DMSC-1/2017 | TS-1, C6/36-3 | Serum | 1.79, 0.66, 0.90–3.90 | MT377491 |
| MU-DMSC-2/2017 | TS-1, C6/36-4 | Serum | 1.25, 0.47, 0.16–2.02 | MT377493 |
| MU-DMSC-3/2017 | TS-1, C6/36-3 | Serum | 1.34, 0.41, 0.44–1.94 | MT377494 |
| MU-DMSC-4/2017 | TS-1, C6/36-4 | Serum | 1.34, 0.34, 0.53–1.86 | MT377495 |
| MU-DMSC-5/2017 | TS-1, C6/36-3 | Serum | 1.01, 0.57, 0.45–2.24 | MT377501 |

## Plaque formation of ZIKV isolates

The ZIKV isolates' capability to form plaques was determined in Vero cell monolayers maintained in 1.5%CMC semi-solid media. All 11 ZIKV isolates formed plaques of various sizes. Moreover, small plaques were found mixing with regular size plaques, suggesting the presence of quasi-species in the virus population of a ZIKV isolate at early passage. The average plaque size of these virus isolates varied from 0.79 to 4.4 mm. (Table 1). The MUMT-1/2016 ZIKV isolate gave the largest plaque size, and the MU-DMSC-6/2016 viral isolate gave the smallest. Examples of plaques of various sizes from three ZIKV isolates are shown in Fig 1, and from eight more isolates in S1 Fig.

## Virus titration

Quantification of the numbers of infectious viruses in ZIKV stocks was carried out by plaque and FFU assays. The results obtained from the two methods were comparable. The titers of most of the ZIKV isolate stocks were $10^6$–$10^7$ PFU or FFU per ml. Overall, the viral titers obtained by plaque and FFU assays differed by 0.01–0.51 log10.

Quantification of the viral genomes was performed by real-time and digital droplet RT-PCR. Based on the real-time RT-PCR and the PAHO protocol, Ct values of 12–14 were

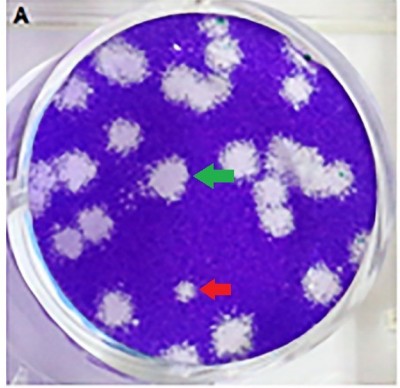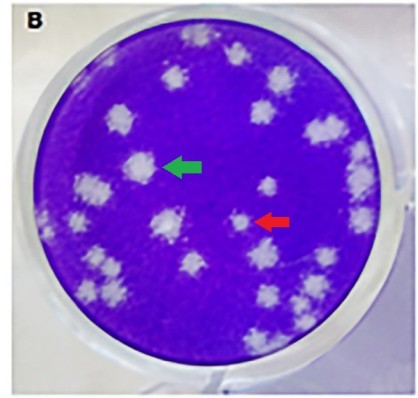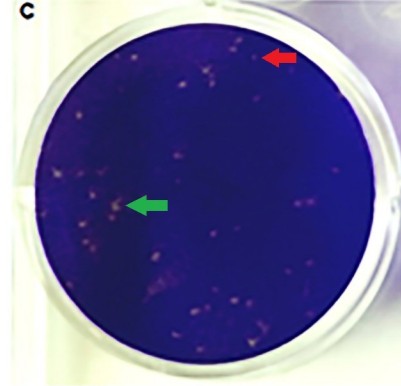

**Fig 1.** The difference in plaque sizes and mixed population (the bigger plaque shown by green arrow and smaller plaque shown by red arrow) of ZIKV isolates; (A) MUMT-1/2016 (2.37–4.40 mm.), (B) MU-DMSC-1/2017 (0.90–3.90 mm.) and (C) MU-DMSC-6/2016 (0.17–2.24 mm).

**Table 2. Comparison of Zika viral titer between Plaque Forming Unit (PFU), Focus Forming Unit (FFU), real time RT-PCR (Ct) and droplet digital RT-PCR (copy number/ml).**

| Virus name | Cell-based | | Molecular-based | |
|---|---|---|---|---|
| | PFU | FFU | Ct | Copy number/ml |
| MUMT-1/2016 | $5.60 \times 10^7$ | $2.40 \times 10^7$ | 13.214 | $9.47 \times 10^8$ |
| MU-DMSC-2/2016 | $3.60 \times 10^6$ | $1.12 \times 10^6$ | 14.043 | $8.93 \times 10^8$ |
| MU-DMSC-3/2016 | $1.04 \times 10^6$ | $5.53 \times 10^5$ | 13.416 | $9.61 \times 10^8$ |
| MU-DMSC-4/2016 | $1.26 \times 10^7$ | $9.93 \times 10^6$ | 12.621 | $1.23 \times 10^9$ |
| MU-DMSC-5/2016 | $1.17 \times 10^6$ | $1.40 \times 10^6$ | 13.208 | $9.70 \times 10^8$ |
| MU-DMSC-6/2016 | $9.88 \times 10^5$ | $1.20 \times 10^6$ | 12.339 | $1.42 \times 10^9$ |
| MU-DMSC-1/2017 | $1.83 \times 10^7$ | $1.26 \times 10^7$ | 13.660 | $5.99 \times 10^8$ |
| MU-DMSC-2/2017 | $1.48 \times 10^6$ | $1.13 \times 10^6$ | 12.603 | $1.09 \times 10^9$ |
| MU-DMSC-3/2017 | $4.00 \times 10^6$ | $1.53 \times 10^6$ | 13.247 | $9.66 \times 10^8$ |
| MU-DMSC-4/2017 | $1.24 \times 10^6$ | $1.46 \times 10^6$ | 13.731 | $9.29 \times 10^8$ |
| MU-DMSC-5/2017 | $1.00 \times 10^6$ | $1.03 \times 10^6$ | 14.311 | $4.99 \times 10^8$ |

obtained for all virus isolates. The Ct value was inversely correlated with the number of the viral genomes, but the exact number was unknown. On the other hand, the digital droplet RT-PCR determined the copy number of the viral genomes in the ZIKV stocks. Using digital PCR, the number of ZIKV genomes of all 11 viral isolate stocks lay between $10^8$ and $10^9$ copies/ml. These values were 1.13–3.16 log10 greater than the respective PFU and 1.60–3.24 log10 greater than the FFU. The values of the virus titers obtained by all 4 methods were compared, as shown in Table 2.

## Discussion

Virus isolation was the gold standard method for viral disease diagnosis for many years until it was replaced by molecular techniques which identify viruses at the level of type, subtype and clade. Nevertheless, viral isolates are essential for studying their biology, e.g., morphology, size and structure of the viral particles, pathogenesis, drug sensitivity, and for vaccine development and evaluation.

Previous investigators showed that the dengue sera caused cell toxicity when directly inoculated onto the C6/36 cell monolayers during virus isolation [16]. The isolation rate of dengue viruses was increased using a mosquito inoculation technique [17]. We found that serum samples from ZIKV patients coagulated in the Vero or C6/36 culture supernatants, and failed to isolate ZIKV from a number of these specimens. Therefore, we employed the mosquito inoculation technique to solve the problem of serum toxicity, expecting to increase the ZIKV isolation rate.

We intrathoracically inoculated ZIKV genome-positive specimens into *Tx. splendens*. After 10 days of incubation, the inoculated mosquitoes were pooled and subjected to RT-PCR for detection of ZIKV genomes. Mosquito suspensions found to be RT-PCR positive were inoculated into the C6/36 cell cultures for three blinded passages. From 270 pools of mosquito suspensions, we obtained 30 real-time RT-PCR positive pools and a final yield of 11 ZIKV isolates. We obtained only one (0.6%) ZIKV isolate from 171 urine samples, 8 (8.7%) from 92 serum samples and 2 from 7 kinds of autopsy tissue from an abortive fetus. ZIKV is not stable in the urine. It degrades within 10 days, even when stored at -80˚C [18]. Our study employed archival urine samples that were stored at -80˚C for longer than one year. This likely explains our low ZIKV isolation rate with the urine specimens. The isolation rate was better with archival serum specimens. Moreover, we succeeded in isolating ZIKV by directly inoculating

autopsy tissue suspensions onto Vero or C6/36 cell monolayers (unpublished data). Our colleague obtained only one ZIKV isolate out of 368 RT-PCR positive serum samples directly inoculated onto Vero cell cultures [19]. Of note, the plasma specimens may be superior to serum specimens. Previous investigators obtained 4 (9.5%) ZIKV isolates from 42 plasma samples after 3 passages onto Vero or C6/36 cells. Their yield was even higher when the plasma samples were inoculated onto monocyte-derived macrophage cultures for one passage, followed by viral expansion in Vero or C6/36 cells [20]. Unfortunately, we did not access to plasma specimens.

Plaque assay is a classical cell-based method to quantify the number of infectious viruses. Nevertheless, some viral isolates do not form plaques or form plaque with ill-defined morphology. In this study, the plaque sizes of ZIKV isolates varied from 0.79 to 4.40 mm. Mixed plaque sizes were observed with a single ZIKV isolate, suggesting the presence of a mixed virus population. Nevertheless, using the Sanger method for nucleotide sequencing, we did not see any double peaks in the chromatograms, a marker of a virus quasi-species.

Moreover, difference in plaque sizes across the viral isolates, particularly the larger plaque size of an isolate from the brain stem, may be related to the viral virulence. We are analyzing the virulence determinants and phylogenetic relationship of our genomic sequences against the others. Plaque assay is laborious, takes time, and consumes large volumes of reagents. The FFU assay was developed as an alternative cell-based method to the plaque assay. It relies on immunostaining techniques with tagged antibodies to demonstrate intracellular viral proteins [21]. The assay is as accurate as the plaque assay. Compared to the plaque assay, the FFU assay is faster, consumes less reagents, and has higher throughput. The FFU assay is run on 96-well plates, whereas the plaque assay employs 6 well- or 12 well- plates [22]. Our immunostaining in the FFU assay employed the 4G2 monoclonal antibody targeting the dengue-2 viral envelope and broadly reacts across flaviviruses [23]. We showed that the titers of each ZIKV isolate, as determined by plaque and FFU assays, were comparable. Unfortunately, the foci of infected cells from the FFU assay were too tiny to reveal the presence of virus quasi-species.

We also determined the amount of ZIKV genomes by molecular techniques. Real-time RT-PCR semi-quantitates the viral genomes as shown by the cycle threshold (Ct). This Ct value is inversely correlated with the copy numbers of the viral genome. Actual copy numbers of the target genes can be determined if the run is conducted along with the standard viral RNA and a calibration curve. In contrast, digital RT-PCR (based on measurement of a large number of positive micro-reactions in oil droplets) can determine the copy numbers of target genes without viral RNA standard and calibration curve [13, 24].

In our virus suspensions, the genome copy numbers measured by the molecular techniques were usually higher than the infectious virus titers determined by PFU or FFU assays. This was likely due to the ability of the molecular techniques to measure excess genomes that did not assemble into the virus particles [25] or to defective interfering particles that arose during viral replication [26, 27]. The molecular-based quantification methods require high-technology machines and expensive reagents that are not required by the infectivity-based assays. This study demonstrated that the ZIKV quantitation by the PFU assay, FFU assay, real-time RT-PCR, and digital RT-PCR were well correlated. The selection of the titration method should rely on the objectives of the specific research study. For example, molecular-based assays are suitable for the studies that need to know the amount of the viral genomes or monitor the viral response to drug treatment. In contrast, infectivity-based assays are suitable for the studies on serodiagnosis or determining virus reduction by anti-viral agents using neutralization assay.

## Supporting information

**S1 Fig.** The plaque sizes of 8 ZIKV isolates; (A) MU-DMSC-2/2016, (B) MU-DMSC-3/2016, (C) MU-DMSC-4/2016, (D) MU-DMSC-5/2016, (E) MU-DMSC-2/2017, (F) MU-DMSC-3/2017, (G) MU-DMSC-4/2017 and (H) MU-DMSC-5/2017.
(TIF)

## Acknowledgments

We thank Tipsuda Chanmanee, Nattakan Thinpan and Anusara Jitsatja, Faculty of Medical Technology, Mahidol University, for their laboratory assistance. We also thank Dusit Noree and Prapanit Wansopa, the National Institute of Health, Department of Medical Sciences, Ministry of Public Health, for the mosquito cultivation. We also thank Dr.Arthur Brown for English editing.

## Author Contributions

**Conceptualization:** Worawat Dangsagul, Pilaipan Puthavathana.

**Funding acquisition:** Pilaipan Puthavathana.

**Investigation:** Worawat Dangsagul, Don Changsom.

**Methodology:** Worawat Dangsagul, Pilaipan Puthavathana.

**Resources:** Kriengsak Ruchusatsawat, Apiwat Tawatsin, Sukontip Putchakarn, Chayawat Phatihattakorn, Prasert Auewarakul.

**Supervision:** Pilaipan Puthavathana.

**Writing – original draft:** Worawat Dangsagul, Pirom Noisumdaeng, Pilaipan Puthavathana.

**Writing – review & editing:** Worawat Dangsagul, Pirom Noisumdaeng, Pilaipan Puthavathana.

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
