## [Decision Letter · Decision Letter 0]

6 May 2021

PONE-D-21-11564

Zika virus isolation, propagation, and quantification using multiple methods

PLOS ONE

Dear Prof. Puthavathana

Thank you for submitting your manuscript to PLOS ONE. After careful consideration, we feel that it has merit but does not fully meet PLOS ONE’s publication criteria as it currently stands. Therefore, we invite you to submit a revised version of the manuscript that addresses the points raised during the review process.

It would be critically important addressing the issue raised by the reviewer about the genotype of the ZIKV isolates exhibiting different plaque phenotypes. In addition, the reviewer cited a number of minor issues that you need to adequately address in the revised version of the paper.

We look forward to receiving your revised manuscript.

Kind regards,

Juan Carlos de la Torre, Ph.D.

Academic Editor

PLOS ONE

Journal Requirements:

PLOS requires an ORCID iD for the corresponding author in Editorial Manager on papers submitted after December 6th, 2016. Please ensure that you have an ORCID iD and that it is validated in Editorial Manager. To do this, go to ‘Update my Information’ (in the upper left-hand corner of the main menu), and click on the Fetch/Validate link next to the ORCID field. This will take you to the ORCID site and allow you to create a new iD or authenticate a pre-existing iD in Editorial Manager. Please see the following video for instructions on linking an ORCID iD to your Editorial Manager account: https://www.youtube.com/watch?v=_xcclfuvtxQ

Please provide additional details regarding participant consent. In the ethics statement in the Methods and online submission information, please ensure that you have specified (1) whether consent was informed and (2) what type you obtained (for instance, written or verbal, and if verbal, how it was documented and witnessed). If your study included minors, state whether you obtained consent from parents or guardians. If the need for consent was waived by the ethics committee, please include this information.

Reviewers' comments:

Reviewer's Responses to Questions

**Comments to the Author**

1. Is the manuscript technically sound, and do the data support the conclusions?

Reviewer #1: Partly

2. Has the statistical analysis been performed appropriately and rigorously? 

Reviewer #1: N/A

3. Have the authors made all data underlying the findings in their manuscript fully available?

Reviewer #1: Yes

4. Is the manuscript presented in an intelligible fashion and written in standard English?

Reviewer #1: No

5. Review Comments to the Author

Reviewer #1: In this manuscript “Zika virus isolation, propagation, and quantification using multiple methods” by Dangsagul et al., the authors attempted to isolate Zika virus (ZIKV) from a total of 270 specimens (92 sera, 171 urines, and 7 autopsy tissue samples) by intrathoracic inoculation of samples into Toxorhynchitis splendens mosquitoes, followed by three passages in C6/36 mosquito cells. From these 270 samples, the authors were able to isolate ZIKV from 11 samples. Isolated ZIKV were titrated by plaque (plaque forming units, PFU) and immunofluorescence (focus forming units) assays in Vero cells and viral genome copies were determined by real time RT-PCR and digital RT-PCR. The manuscripts demonstrate the feasibility of isolate ZIKV using inoculation of Toxorhynchitis splendens mosquitoes followed by passage in C6/36 mosquito cells, and that plaque and immunofluorescence assays results in similar viral titers while quantification by real-time or digital droplet RT-PCR were higher than those obtained using plaque or immunofluorescence assays.

Overall, this is a descriptive manuscript that could probably fit better in a methodology journal since provide with technical approaches for isolation and titration of ZIKV. Notably, although the authors have been able to isolate and observe differences in the plaque phenotype of the isolated ZIKV, genome sequence of the viral genomes have not been provided. The manuscript will improve if the authors provide with the sequence analysis of the different isolated ZIKV that could explain the differences in the plaque phenotype. This is particularly important because the virus with the biggest plaque phenotype (MUMT-1/2016) was isolated from a brain sample, contrary to the other ZIKV that were isolated from either urine or serum samples.

There are also some minor concerns:

1) The authors could provide with plaque assays for the 11 isolated ZIKV (Table 1) rather than a selected set of 3 of them (Figure 1).

2) The authors indicate that some of the ZIKV isolates presented with viruses with different plaque sizes, suggesting the presence of mix population of viruses. However, plaque assays shown in Figure 1 suggest a similar plaque phenotype in the three isolated ZIKV MUM-1/2016 (A), MU-DMSC-1/2017 (B) and MU-DMSC-6/2016.

3) The authors should revise the manuscript to provide with some missing information (e.g. source of Vero cell lines) and to keep consistency with the nomenclature (e.g. ZIKV vs. Zika virus).

4) Line 219 indicate Table 2 but I think the authors refer to Table 1.

6. PLOS authors have the option to publish the peer review history of their article (what does this mean?). If published, this will include your full peer review and any attached files.

Reviewer #1: No

---

## [Author Response · Author response to Decision Letter 0]

23 Jun 2021

Response to Reviewer

PONE-D-21-11564

Article title: Zika virus isolation, propagation, and quantification using multiple methods

Reviewers' comments:

Reviewer's Responses to Questions

Comments to the Author

1. Is the manuscript technically sound, and do the data support the conclusions?

Reviewer #1: Partly

2. Has the statistical analysis been performed appropriately and rigorously?

Reviewer #1: N/A

3. Have the authors made all data underlying the findings in their manuscript fully available?

Reviewer #1: Yes

4. Is the manuscript presented in an intelligible fashion and written in standard English?

Reviewer #1: No

Response: We have Dr. Arthur Brown, a native English speaker, editing our language in the revised manuscript as shown in acknowledgement.

5. Review Comments to the Author

Reviewer #1: In this manuscript “Zika virus isolation, propagation, and quantification using multiple methods” by Dangsagul et al., the authors attempted to isolate Zika virus (ZIKV) from a total of 270 specimens (92 sera, 171 urines, and 7 autopsy tissue samples) by intrathoracic inoculation of samples into Toxorhynchitis splendens mosquitoes, followed by three passages in C6/36 mosquito cells. From these 270 samples, the authors were able to isolate ZIKV from 11 samples. Isolated ZIKV were titrated by plaque (plaque forming units, PFU) and immunofluorescence (focus forming units) assays in Vero cells and viral genome copies were determined by real time RT-PCR and digital RT-PCR. The manuscripts demonstrate the feasibility of isolate ZIKV using inoculation of Toxorhynchitis splendens mosquitoes followed by passage in C6/36 mosquito cells, and that plaque and immunofluorescence assays results in similar viral titers while quantification by real-time or digital droplet RT-PCR were higher than those obtained using plaque or immunofluorescence assays.

Overall, this is a descriptive manuscript that could probably fit better in a methodology journal since provide with technical approaches for isolation and titration of ZIKV. Notably, although the authors have been able to isolate and observe differences in the plaque phenotype of the isolated ZIKV, genome sequence of the viral genomes have not been provided. The manuscript will improve if the authors provide with the sequence analysis of the different isolated ZIKV that could explain the differences in the plaque phenotype. This is particularly important because the virus with the biggest plaque phenotype (MUMT-1/2016) was isolated from a brain sample, contrary to the other ZIKV that were isolated from either urine or serum samples.

Response: We conducted complete genome sequencing of these 11 ZIKV isolates by the Sanger method and deposited the sequencing data in the GenBank database. The sequencing data will be released in June 2021 using the accession numbers as shown in Table 1. The information on molecular characterization of these 11 viruses together with the other ZIKV isolates in Thailand is shown in the other manuscript under reviewing process by the other publisher. In response to this important comment from the reviewer, we add more information about the genetics of these 11 ZIKV isolates in the Materials and Methods, and the Discussion of the revised manuscript as follows.

Line Nos. 201-204: 

Genomic characteristics of the ZIKV isolates 

Complete genome sequencing by the Sanger method showed that all 11 ZIKV isolates belonged to the Asian lineage. These sequences can be retrieved from the GenBank database using the accession numbers shown in Table 1. 

Line Nos. 276-281: 

Nevertheless, using the Sanger method for nucleotide sequencing, we did not see any double peaks in the chromatograms, a marker of a virus quasi-species. Moreover, difference in plaque sizes across the viral isolates, particularly the larger plaque size of an isolate from the brain stem, may be related to the viral virulence. We are analyzing the virulence determinants and phylogenetic relationship of our genomic sequences against the others. 

There are also some minor concerns:

1) The authors could provide with plaque assays for the 11 isolated ZIKV (Table 1) rather than a selected set of 3 of them (Figure 1).

Response: We have included the plaque characteristics of 8 more ZIKV isolates in the S1 Fig. We have included this information in Line No. 223.

2) The authors indicate that some of the ZIKV isolates presented with viruses with different plaque sizes, suggesting the presence of mix population of viruses. However, plaque assays shown in Figure 1 suggest a similar plaque phenotype in the three isolated ZIKV MUM-1/2016 (A), MU-DMSC-1/2017 (B) and MU-DMSC-6/2016.

Response: The three ZIKV isolates mentioned displayed different plaque sizes, but the limitation of photograph resolution may have made the small plaque sizes not clearly visible.

We put the arrows to make the pictures clearer.

3) The authors should revise the manuscript to provide with some missing information (e.g. source of Vero cell lines) and to keep consistency with the nomenclature (e.g. ZIKV vs. Zika virus).

Response: The C6/36 cell line (ATCC CRL-1660) and Vero cell line (ATCC CCL-81) were used. Please see Line no.112, and 149, respectively. We replace the nomenclature of Zika virus with ZIKV.

4) Line 219 indicate Table 2 but I think the authors refer to Table 1.

Response: Thank you. We change “Table 2” in line 221 to “Table 1”.

---

## [Editor Report · Decision Letter 1]

28 Jun 2021

PONE-D-21-11564R1

Zika virus isolation, propagation, and quantification using multiple methods

PLOS ONE

Dear Dr. Puthavathana,

Thank you for submitting your manuscript to PLOS ONE. After careful consideration of your revised version of the paper, we concluded that still needs a minor revision to thoroughly address the issue raised by the reviewer about the different plaque size phenotypes (see comments for Author). Therefore, we invite you to submit a revised version of the manuscript that addresses this minor issue. 

We look forward to receiving your revised manuscript.

Kind regards,

Juan Carlos de la Torre, Ph.D.

Academic Editor

PLOS ONE

Journal Requirements:

Additional Editor Comments (if provided):

This revised version of a paper by Dangsagul and colleagues has addressed the main concerns raised by the reviewer of the original submission, with exception of the question raised by the reviewers about the plaque size phenotype.

The response provided by the authors is not satisfactory. The authors need to provide additional information about the characterization of the plaque size phenotype by confirming a stable plaque size phenotype of plaque purified viral populations exhibiting different plaque size shown in figure 1 (red and green arrows).

---

## [Author Response · Author response to Decision Letter 1]

10 Jul 2021

PONE-D-21-11564R1 - [EMID:d7f4c12d9bb6967e]

Article title: Zika virus isolation, propagation, and quantification using multiple methods

Additional Editor Comments (if provided): 

This revised version of a paper by Dangsagul and colleagues has addressed the main concerns raised by the reviewer of the original submission, with exception of the question raised by the reviewers about the plaque size phenotype.

The response provided by the authors is not satisfactory. The authors need to provide additional information about the characterization of the plaque size phenotype by confirming a stable plaque size phenotype of plaque purified viral populations exhibiting different plaque size shown in figure 1 (red and green arrows).

Response to additional comment

Plaque variants or the mixed plaque size phenotype are common in several virus isolates of early passages. This phenotype may be due to the difference in fitness of the individual infectious virus to grow in new hosts. Usually, the virus with a larger plaque size replicates faster and will overgrow and replace the smaller plaque size after sub-passaging. Consequently, the virus preparation will contain a homogeneous population of larger plaque size. 

Virologists performed plaque purification and characterization for specific purposes, such as to study the viral virulence determinants or when the purified virus population is essential for vaccine development or molecular cloning. Plaque purification and characterization by themselves are sufficient to serve as the specific aims in an article, as exampled below. 

1. Kato F, Tajima S, Nakayama E, Kawai Y, Taniguchi S, Shibasaki K, et al. Characterization of large and small-plaque variants in the Zika virus clinical isolate ZIKV/Hu/S36/Chiba/2016. Sci Rep. 2017;7(1):16160. https://doi.org/10.1038/s41598-017-16475-2.

2. Mandary MB, Masomian M, Ong SK, Poh CL. Characterization of plaque variants and the involvement of quasi-species in a population of EV-A71. Viruses. 2020;12(6):651. https://doi.org/10.3390/v12060651.

3. Jia Y, Moudy RM, Dupuis AP, Ngo KA, Maffei JG, Jerzak GV, et al. Characterization of a small plaque variant of West Nile virus isolated in New York in 2000. Virology. 2007;367(2):339-47. https://doi.org/10.1016/j.virol.2007.06.008. 

Nevertheless, the objectives of our manuscript are to isolate Zika viruses from clinical specimens and determine the virus titers using multiple techniques: plaque assay, focus forming unit assay, droplet digital RT-PCR, real time RT-PCR. The plaque assay demonstrated that our Zika virus isolates were consisted of the population with mixed plaque sizes. Individual plaque, regardless of size or virulence was count as one infectious unit. Therefore, the objective of our manuscript on virus titration was fulfilled. Plaque purification and purification can take a year to finish, and the result gained does not add up much information for this manuscript. The technique should be carried out to serve a specific purpose that is more worthwhile than the virus quantification. 

According to the explanation described above, we do not change the revised version of our manuscript. Please reconsider.

---

## [Editor Report · Decision Letter 2]

14 Jul 2021

Zika virus isolation, propagation, and quantification using multiple methods

PONE-D-21-11564R2

Dear Dr. Puthavathana,

We’re pleased to inform you that your manuscript has been judged scientifically suitable for publication and will be formally accepted for publication once it meets all outstanding technical requirements.

Kind regards,

Juan Carlos de la Torre, Ph.D.

Academic Editor

PLOS ONE

Additional Editor Comments (optional):

In this second revision of their paper, the authors have addressed the comment raised by the reviewer about the plaque size phenotype.

Although the response is far from satisfactory, the plaque size phenotype represents only a minor component of the paper and therefore it has a limited impact on the overall content of the paper.
---

## [Editor Report · Acceptance letter]

21 Jul 2021

PONE-D-21-11564R2 

Zika virus isolation, propagation, and quantification using multiple methods 

Dear Dr. Puthavathana:

I'm pleased to inform you that your manuscript has been deemed suitable for publication in PLOS ONE. Congratulations! Your manuscript is now with our production department. 

Kind regards, 

on behalf of

Dr. Juan Carlos de la Torre 

Academic Editor

PLOS ONE